# TEACHING LANGUAGE MODEL TO ACT EFFICIENTLY

## ABSTRACT

Tool-integrated reasoning (TIR) augments large language models (LLMs) with the ability to invoke external tools during long-form reasoning, such as search engines and code interpreters, to solve tasks beyond the capabilities of internal reasoning. While reinforcement learning (RL) has shown promise in training such agents, most of existing approaches typically optimize only for final correctness without considering the efficiency or necessity of external tool use. This often leads to excessive tool calling, leading to increased computational costs and additional latency, and may also shift reliance toward external tools rather than the model's own reasoning – a phenomenon referred to as *cognitive offloading*. To this end, we propose Optimized Tool Call-controlled Policy Optimization (OTC-PO), a simple yet effective RL-based framework that encourages models to produce accurate answers with less tool calls. Our method introduces a tool-integrated reward that jointly considers answer correctness and corresponding tool use behavior of model to reach that answer. To validate the effectiveness, we introduce the additional metric of *tool productivity*, defined as the ratio between the number of correct answers and the total number of tool calls across all test cases. This metric reflects how efficiently and effectively tool usage contributes to successful task completion, with higher values indicating more productive external tool calls with the help of internal reasoning. We then instantiate this framework within both Proximal Policy Optimization (PPO) and Group Relative Preference Optimization (GRPO), resulting in OTC-PPO and OTC-GRPO. Experiments with Qwen-2.5 and Qwen-Math across multiple QA benchmarks show that our approach reduces tool calls by up to 68.3% and improves tool productivity by up to 215.4%, while maintaining comparable answer accuracy, especially for the larger models.

## 1 INTRODUCTION

Large language models (LLMs) have shown strong reasoning capabilities, particularly when trained with reinforcement learning (RL) using simple outcome rule-based rewards (Team, 2024; 2025). Building on this progress, recent work on tool-integrated reasoning (TIR) (Gou et al., 2024; Wang et al., 2024a) enables LLMs to interact with external tools such as search engines (Jin et al., 2025) and code interpreters (Li et al., 2025b) using similar reward design. By combining internal reasoning with access to various external tools, TIR substantially broadens the problem-solving scope of LLMs, allowing them to tackle tasks that exceed the limitations of parametric knowledge alone.

However, most existing RL approaches for TIR focus solely on final answer correctness, without explicitly accounting for the efficiency or necessity of tool use (Jin et al., 2025; Feng et al., 2025). As a result, models often rely excessively on external calls (Qian et al., 2025b). This behavior raises several concerns: (i) higher computational and infrastructure costs due to unnecessary tool invocations (Figure 2 left); (ii) longer inference latency due to lengthy observation returned by the tool (Figure 2 middle); and (iii) a tendency to rely on external tools instead of leveraging the model's own reasoning, a phenomenon known as *cognitive offloading* (Risko & Gilbert, 2016). For example, recent baselines such as Search-R1 (Jin et al., 2025) frequently invoke tools even in cases where the question could be answered directly, missing the opportunity to leverage internal reasoning (§ 5.3).

Based on these observations, we argue that effective agentic LLMs should not only produce correct answers but also use tools judiciously. In practice, different tasks and models may require different levels of external assistance. For some questions, the best strategy involves no tool usage at all, while others may benefit from one or more targeted interactions. This motivates the need for training

methods that optimize both effectiveness (i.e., answer correctness) and efficiency (i.e., appropriate tool call). We aim to develop such model based on a foundational assumption: for *each question* and *each model*, there exists an optimal number of tool calls, defined as the minimal number required for the model to arrive at a correct answer (i.e., the minimum tool calls among correct trajectories practically). Crucially, this optimal number can vary depending on both the capabilities of models and the complexity of questions. Therefore, it is difficult to create supervised fine-tuning data or to design a single prompting strategy that generalizes across diverse models and tasks. In contrast, RL offers a promising alternative, as it allows models to adapt their tool-call behavior based on their own experience or interactions (Silver & Sutton, 2025).

We then propose Optimized Tool Calls controlled Policy Optimization (OTC-PO), a simple yet effective RL-based method that enforces models to minimize the number of tool calls required to reach a correct solution. Specifically, we introduce a tool-integrated reward that modulates traditional outcome reward signals, such as correctness, with a scaling *coefficient* reflecting tool efficiency. This encourages the model to prefer trajectories that reach correct answers with fewer tool calls. Therefore, it shifts the optimization objective from correctness alone to tool productivity, defined as the ratio between task benefit (e.g., answer accuracy) and tool usage cost (e.g., number of tool calls). In addition, OTC-PO is lightweight and plug-and-play, requiring only minimal changes to standard RL pipelines, making it easy to adopt in existing systems. We conduct comprehensive experiments on two widely used tools: search and code based on several LLMs, Qwen-2.5-3B/7B-Base and Qwen2.5-Math-1.5B/7B-Base (Qwen et al., 2025), and derive models with a substantial tool calls reduction in trajectory while approximately maintaining accuracy with strong baselines. In summary, the key contributions are as follows.

- We re-frame tool-integrated reasoning as jointly optimizing correctness and efficiency, and introduce tool productivity, the number of correct answers per tool call, as a metric to capture this trade-off. We additionally identify the cognitive offloading phenomenon in TIR.

- We propose a simple, faster, and generalizable OTC-PO algorithm to encourage the model to use fewer tool calls to solve the problem and therefore maximize tool productivity. We note that it is compatible with various RL algorithms and can be easily implemented.

- We implement OTC-PPO and OTC-GRPO as two typical methods without losing the adaptability and generalization based on PPO (Schulman et al., 2017) and GRPO (Shao et al., 2024) algorithms, and the experimental results on several benchmarks and baselines demonstrate significant reductions in tool call cost while preserving most of the accuracy in both in-domain and out-of-domain evaluation.

## 2 RELATED WORK

**Tool Utilization for LLMs.** Teaching LLMs to use tools enables them to interact with external environments while overcoming several inherent limitations such as restricted access to up-to-date or domain-specific knowledge and poor mathmatical operation capabilities. There are three major methods which can achieve this goal: 1) prompting engineering (Qian et al., 2023; Wang et al., 2024b; Qian et al., 2024b; Yuan et al., 2024; Qian et al., 2024a; Wang et al., 2025a), which guides the model's behavior through carefully designed input templates or few-shot examples without modifying model weights; 2) supervised finetuning on tool-integrated reasoning datasets (Schick et al., 2023; Qin et al., 2023; Gou et al., 2024; Qian et al., 2024c; 2025b), where the model learns from annotated trajectories that demonstrate when and how to use tools in context; and 3) reinforcement learning (Jin et al., 2025; Li et al., 2025b; Feng et al., 2025; Qian et al., 2025a), which allows the model to directly learn tool-use strategies by interacting with an environment and optimizing long-term rewards, enabling more adaptive and goal-directed behaviors. Some previous methods combine SFT and RL with collected tool-use trajectories and preferences (Chen et al., 2025). While prompting and supervised fine-tuning have shown promising results, they rely heavily on expert-designed prompts and tool-integrated annotations, which limits their scalability and generalization. As a result, recent efforts have shifted toward reinforcement learning, which demonstrates improved performance and generalization through simple rule-based rewards.

**Tool-integrated Reasoning Efficiency.** Only a few studies from prompting engineering and supervised fine-tuning attention on tool-integrated reasoning efficiency issues in terms of the cost of

tool usages during the reasoning (Wang et al., 2025a; Shen et al., 2024; Qian et al., 2025b). In detail, (Wang et al., 2025a) first propose a prompting-based framework: Self Divide-and-Conquer (SelfDC) to leverage the self-aware confidence score of LLMs to decide whether or not need to call tools during reasoning, achieving better trade-off between effectiveness and efficiency in the context of RAG. Furthermore, several works follow this direction and explore more boarder applications and larger tool spaces (Qian et al., 2025b; Shen et al., 2024; Li et al., 2025a). For example, SMART (Qian et al., 2025b) collect the well-designed dataset to finetune the model to only call tools when the knowledge is outside the inherent parametric knowledge of LLMs. Despite these advancements, most existing approaches still rely on complex prompt engineering or annotated datasets, which hinders their adaptability and scalability to new scenarios and benchmarks. In contrast, the efficiency of tool-integrated reasoning within reinforcement learning frameworks remains largely underexplored.

## 3 METHODOLOGY

In this section, we first provide a formal definition of task considering the both effectiveness and efficiency of tool-integrated reasoning, followed by general RL framework and then our proposed Optimized Tool Call-controlled Policy Optimization (OTC-PO).

### 3.1 TASK DEFINITION

Given a question $q$, and an environment $\mathcal{E}$ that provides access to a set of tools $\mathcal{T} = \{t_0, t_1, ...t_n\}$, the language model $\mathcal{M}$ can *optionally* interact with the environment by calling specific tools in $\mathcal{T}$, obtaining the corresponding tool results from $\mathcal{E}$, and iteratively repeating this processing until the final answer is driven. Without losing generalization, the tool-integrated reasoning trajectory $\tau_k$ at step $k$ is defined as follows:

$$\tau_k = (r_1, tc_1, o_1), (r_2, tc_2, o_2), ...(r_k, tc_k, o_k), \tag{1}$$

where $r_i, tc_i, o_i$ denotes the reasoning, tool call and returned observation respectively. Importantly, we also account for reasoning steps that do not involve tool usage. Suppose the step $p$ does not need to call tools, then the $tc_p$ and $o_p$ become empty string, the reasoning content $r_p$ can either be merged with the subsequent reasoning step $r_{p+1}$ to form the new $r_{p+1}^*$, or, if $p$ is the last step [1], be used directly to derive the final answer. The objective of the task is to generate the correct answer $\hat{a}$ with the minimal cost of the full trajectory $\tau$ as follows:

$$\arg\min_{\tau} \ \text{Cost}(\tau) \quad \text{subject to} \quad \mathcal{M}(q, \tau) = \hat{a}, \tag{2}$$

Here the cost is measured as the number of tool calls within the trajectory $\tau$ considering its simplicity and generalization. Thus the model is encouraged to not only generate correct answer but also minimize the cost. We emphasize that this revised objective highlights an overlooked dimension of TIR to balance correctness with the tool use behavior of models.

### 3.2 TOOL-INTEGRATED REINFORCEMENT LEARNING

Inspired by recent success to use RL for better reasoning in LLMs, several efforts try to extend RL to tool-integrated reasoning with the objective functions can be defined as follows:

$$\max_{\pi_\theta} \mathbb{E}_{q \sim \mathcal{D}, \, y \sim \pi_\theta(\cdot|q;\mathcal{E})} \left[ r_\phi(q, y) \right] - \beta \, \mathbb{D}_{kl} \left[ \pi_\theta(y \mid q; \mathcal{E}) \, \| \, \pi_{\text{ref}}(y \mid q; \mathcal{E}) \right], \tag{3}$$

where $\pi_\theta, \pi_{ref}$ stand for the policy model and reference model respectively, $r_\phi$ is the reward function and $\mathbb{D}_{kl}$ is the KL-divergence measure. $q$ is the question drawn from the dataset $\mathcal{D}$ and $y$ is the generated output, consisting of the tool-integrated reasoning trajectory $\tau$ and the final answer $a$. To optimize this goal, there are two well-established policy-gradient RL methods: Proximal Policy Optimization (PPO) and Group Relative Policy Optimization (GRPO) in TIR. We provide detailed optimization objective in Appendix C to save space.

---

[1]The last step is practically determined by the predefined maximal tool calls.

Figure 1: An overview of OTC-GRPO Algorithm.

Following the success of DeepSeek-R1 (Team, 2025), most prior predominantly focuses on rule-based outcome correctness rewards (Jin et al., 2025; Li et al., 2025b; Feng et al., 2025) to learn the policy model during training, as illustrated below:

$$r_\phi(q, y) = r_{correct} = 1 \text{ if } a = \hat{a} \text{ else } 0, \tag{4}$$

where $a$ is the extracted predicted answer from the response $y$ and $\hat{a}$ is the ground truth answer.

### 3.3 OTC-PO: Optimized Tool Call-controlled Policy Optimization

In order to consider the tool call behavior, we propose Optimized Tool Call-controlled Policy Optimization (OTC-PO), which can be easily integrated with existing RL algorithms, aiming to preserve general accuracy while significantly reducing the cost of tool interactions. At the heart of OTC-PO is a reward mechanism that ranks the current number of tool calls $m$ relative to the optimal number of tool calls $n$ given the question and model. In most realistic settings, the true optimal number $n$ is unknown in advance. To address this, OTC-PO approximates $n$ by tracking the minimal number of tool calls observed across different trajectories to arrive the correct answer for the same question and model. This approach enables the model to learn efficient tool use without requiring prior knowledge of the optimal tool budget. Furthermore, the framework can be naturally extended to scenarios where the optimal tool call number is known a priori (e.g., $n = 0$ for language-only reasoning). We instantiate OTC-PO in two variants: OTC-PPO and OTC-GRPO, tailored to different underlying RL algorithms.

**OTC-PPO.** Since there is no multiple trajectories for same question $q$ in single optimization step, we design the tool reward according to the number of tool calls $m$ in the current trajectory as follows:

$$r_{tool} = \cos(\frac{m * \pi}{2m + c}) \tag{5}$$

The core idea is to map $m$ to $[0, \pi/2)$ and $c$ is the smooth constant that controls the reward decay rate. The smaller the value, the faster the punishment and the more it encourages less use of tools; the larger the value, the more tolerant it is. In this way, among trajectories, those requiring more tool calls will receive lower rewards than those achieved with fewer tool calls. We note that any function exhibiting such monotonicity is applicable, but we find sin/cos functions is more smooth and easy to train, as also evidence by Yeo et al. (2025). Although we can track the $m$ here for approximation of $n$ in later steps, we empirically find it requires much longer training steps due to poor sampling efficiency in PPO, which is undesirable.

**OTC-GRPO.** We first identify the trajectories $\{\tau^1, \tau^2, ..., \tau^p\}$ leading to correct answer from the group for the question $q$, and get the corresponding number of tool calls for each trajectory: $\mathcal{C} = \{c^1, c^2, ..., c^p\}$, and we can calculate the minimal tool calls $k = \min(\mathcal{C})$, serving as the *local* approximation of optimal tool calls for $q$. Furthermore, we can update $k$ during multiple epochs to approximate the *global* optimal tool calls if the policy model finds the better solution with less than $k$ calls in later iterations. We use $n$ to indicate the approximation of optimal tool calls, and $m$ the number of tool calls in the current trajectory for $q$. Therefore we design the reward as follows:

$$r_{tool} = \begin{cases} 1 & \text{if } f(m,n) = n = 0 \\ \cos(\frac{m*\pi}{2m+c}) & \text{if } n = 0 \\ \sin(\frac{f(m,n)*\pi}{2n}) & \text{otherwise} \end{cases} \qquad f(m,n) = \begin{cases} 0, & \text{if } m = 0 \text{ and } n = 0 \\ m, & \text{if } n = 0 \\ \dfrac{2nm}{m+n}, & \text{otherwise} \end{cases}$$

(6)

where $f(m,n)$ is the mapping function to re-map the $m$ to the range from 0 to $2n$. Briefly, the key here is to assign the highest reward 1 (a.k.a, $\sin(\pi/2)$) when the policy model achieves optimal tool calls $n$, and when actual number of tool calls $m$ deviates from $n$, either exceeding or falling short, the model receives a comparatively lower reward. Thus, we can assign different rewards dynamically according to the different $n$ and $m$ for the question $q$. We provide reward illustration to better motivate our reward design in Appendix E.1.

**Tool-integrated Reward Design.**   Inspired by recent study (Arora & Zanette, 2025), we regard the tool-integrated reward $r_{tool}$ as a *coefficient* of conventional reward function $r_\phi(q,y)$, leading to the final tool-integrated reward function:

$$r_\phi^{tool}(q,y) = \alpha * r_{tool} * r_\phi(q,y) \qquad (7)$$

where $\alpha$ is a hyperparameter that controls the scaling of the tool reward as $r_\phi$ is usually same for correct or wrong answer. Notably, this multiplicative structure ensures that tool efficiency is mainly rewarded when the primary task objective (e.g., answer correctness) is satisfied. For example, when the final answer is incorrect, $r_\phi$ is 0, effectively disabling the influence of $r_{tool}$. This design helps mitigate reward hacking by preventing the policy model from being incentivized to use tools without ultimately producing a correct answer. If the answer is correct, $r_\phi$ is 1 and then $r_{tool}$ begins to dominate the scale of the reward, thus encouraging less tool usage.

It is worth noting that this design offers several practical advantages: i) It is consistent with the task objective, which prioritizes achieving correct solutions while reducing the number of tool calls, with theoretical support provided in Appendix E.2. ii) It helps reduce the risk of reward hacking compared to simple additive reward formulations (e.g., $r_{tool} + r_\phi$), which in our experiments led to unstable behaviors. For instance, we observed cases where the model could achieve higher reward by minimizing tool usage without necessarily producing correct answers. iii) It remains flexible and can be applied to different reward formulations of $r_\phi$, such as $r_\phi = r_{correct}$ or $r_\phi = r_{correct} + r_{format}$, depending on the task requirements, as discussed in Appendix E.3.

## 4 Experiments

### 4.1 Set up

**Datasets and Baselines.**   We mainly follow the Search-R1 (Jin et al., 2025) and ToRL (Li et al., 2025b) setting and use same baselines and datasets for the fair and comprehensive evaluation. Specifically, we use NQ and HotpotQA as training dataset for search, and we use the dataset provided in ToRL for code. We also directly compare our methods with several baselines such as SFT, Base-RL, retrieval-augmented generation baselines (i.e., RAG, IRCoT) and more importantly the Search-R1 and ToRL themselves.

**Evaluation Metrics.**   Besides the exact match (EM) and the average tool calls (TC), we additionally define a new metric, *tool productivity* (TP), which measures the effectiveness and efficiency of tool calls during inference. Specifically, TP is defined as the number of correctly answered questions per unit of tool call: TP = $\frac{\sum_{i=1}^{N} \mathbb{I}\{y_i = \hat{y}_i\}}{\sum_{i=1}^{N} tc_i}$ [2], where $\mathbb{I}$ is the indicator function that equals 1 if the predicted answer $\hat{y}_i$ matches the ground truth $y_i$, and $tc_i$ denotes the number of tool calls used in the $i_{th}$ instance. This metric reflects how efficiently the model converts tool usage into correct answers, capturing both utility and cost in a single measure. While EM provides a standard measure of accuracy, it does not reflect the underlying reasoning cost. Therefore, we consider TP as more

---

[2]This can also be understood as the fraction between benefits and cost.

Table 1: The results of OTC-PO with different baselines in search. The results except Search-R1 are directly copied from original paper (Jin et al., 2025). We highlight the relative improvement of OTC-PO and OTC-GRPO compared with the corresponding variants of Search-R1.

| Models | NQ | | | HotpotQA | | |
|---|---|---|---|---|---|---|
| | EM (↑) | TC (↓) | TP (↑) | EM (↑) | TC (↓) | TP (↑) |
| **Qwen2.5-3B(-Base)** | | | | | | |
| R1-Base | 0.226 | - | - | 0.201 | - | - |
| SFT | 0.249 | - | - | 0.186 | - | - |
| RAG | 0.348 | 1.0 | 0.348 | 0.255 | 1.0 | 0.255 |
| IRCoT | 0.111 | 10.0 | 0.011 | 0.164 | 10.0 | 0.016 |
| Search-R1-PPO | 0.403 | 1.738 | 0.232 | 0.279 | 1.716 | 0.163 |
| Search-R1-GRPO | 0.404 | 1.426 | 0.283 | 0.312 | 1.802 | 0.173 |
| OTC-PPO | 0.355 | 1.010 (▼ 41.9%) | 0.351 (▲ 51.3%) | 0.260 | 1.026 (▼ 40.2%) | 0.253 (▲ 55.2%) |
| OTC-GRPO | 0.444 | 1.008 (▼ 29.3%) | 0.440 (▲ 55.5%) | 0.365 | 1.387 (▼ 23.0%) | 0.263 (▲ 52.0%) |
| **Qwen2.5-7B(-Base)** | | | | | | |
| R1-Base | 0.270 | - | - | 0.242 | - | - |
| SFT | 0.318 | - | - | 0.217 | - | - |
| RAG | 0.349 | 1.0 | 0.349 | 0.299 | 1.0 | 0.299 |
| IRCoT | 0.224 | 9.999 | 0.022 | 0.133 | 9.982 | 0.013 |
| Search-R1-PPO | 0.449 | 3.282 | 0.136 | 0.380 | 3.741 | 0.102 |
| Search-R1-GRPO | 0.399 | 1.697 | 0.235 | 0.341 | 2.109 | 0.162 |
| OTC-PPO | 0.446 | 1.040 (▼ 68.3%) | 0.429 (▲ 215.4%) | 0.383 | 1.464 (▼ 60.9%) | 0.262 (▲ 156.9%) |
| OTC-GRPO | 0.444 | 0.990 (▼ 41.7%) | 0.448 (▲ 90.6%) | 0.366 | 1.005 (▼ 52.3%) | 0.364 (▲ 124.7%) |

Table 2: The results of OTC-PO with different baselines in ToRL (Li et al., 2025b).

| Models | AIME24 | | | AIME25 | | |
|---|---|---|---|---|---|---|
| | EM (↑) | TC (↓) | TP (↑) | EM (↑) | TC (↓) | TP (↑) |
| **Qwen2.5-Math-1.5B(-Base)** | | | | | | |
| Qwen2.5-Math-1.5B-Ins | 10.0 | - | - | 10.0 | - | - |
| Qwen2.5-Math-1.5B-Ins-TIR | 13.3 | 1.1 | 12.1 | 13.3 | 1.4 | 9.5 |
| ToRL-GRPO | 23.3 | 2.2 | 10.6 | 23.3 | 2.3 | 10.1 |
| OTC-GRPO | 20.0 | 1.1 (▼ 50.0%) | 18.2 (▲ 71.7%) | 20.0 | 1.1 (▼ 41.2%) | 18.2 (▲ 80.2%) |
| **Qwen2.5-Math-7B(-Base)** | | | | | | |
| Qwen2.5-Math-7B-Ins | 10.0 | - | - | 16.7 | - | - |
| Qwen2.5-Math-7B-Ins-TIR | 26.7 | 1.6 | 16.4 | 16.7 | 1.4 | 12.2 |
| Base-RL | 33.3 | - | - | 6.7 | - | - |
| ToRL-GRPO | 36.7 | 2.1 | 17.5 | 26.7 | 2.1 | 12.7 |
| OTC-GRPO | 36.7 | 0.7 (▼ 66.7%) | 52.4 (▲ 199.4%) | 23.3 | 0.8 (▼ 61.9%) | 29.1 (▲ 129.1%) |

informative indicators of agentic reasoning efficiency – highlighting not only whether the model can provide correct answer, but how economically it arrives at that correctness.

**Implementation Details.** We re-use the same parameter in Search-R1 (Jin et al., 2025) and ToRL (Li et al., 2025b) respectively. There are only minor modification we make to suit our method: i) We slightly change the template in Search-R1 to tell the model that it only need to call tools when necessary (Appendix D), and we do not change the template in ToRL; ii) We set the maximal number of tool calls $\mathcal{C}$ in ToRL to 3 to better study the effects of our methods when multiple tool calls are allowed and keep it as 4 as in original Search-R1. Moreover, we set $c$ as corresponding max turns or maximal tool limits, $\alpha$ as 1. We conduct our experiments on 8 A100-80G GPU and reproduce the results of Search-R1 and ToRL independently. We implement OTC-GRPO using the global approximation of minimal tool calls since it leads to more stable and better optimization.

## 4.2 MAIN RESULTS

**Search as Tool.** Table 1 shows the results for search-required tasks. There are several key insights can be drawn: i) OTC-PPO achieves significant improvement in terms of TC and TP compared with Search-R1-PPO, resulting in up to a 68.3% reduction in TC and a 215.4% increase in TP, and OTC-GRPO can further reduce the absolute tool calls due to more accurate approximation; ii) It is found that our method will not sacrifice the accuracy too much especially for larger LLMs, as evidenced by our OTC-PO achieves comparable EM score with Search-R1 on Qwen2.5-7B model; iii) Different models have different tool use behaviors on different datasets. If we look at the Search-R1, we can find that 3B model tends to use less tool calls compared with 7B model. This reveals a critical issue: as language models scale up, they may tend to over-rely on external tool calls unless explicitly penalized for such behavior. This not only exacerbates the problem with larger models but also leads to an underutilization of their inherent reasoning capabilities (See $ 5.3). The number

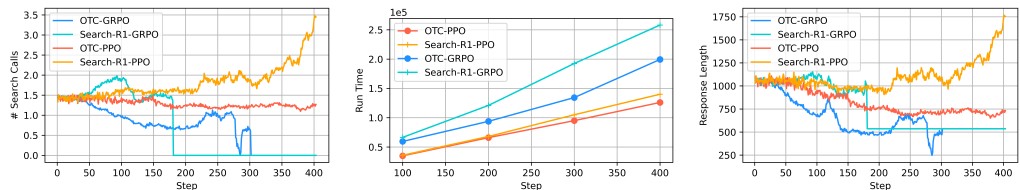

Figure 2: **Left**: Changes of number of search calls during the training; **Middle**: Running time analysis; and **Right**: Changes of response length during the training.

Table 3: Tool behavioral advantage analysis of OTC-PO against Search-R1. ME, LE, MA, LA, AE stands for more efficient, less efficient, more accurate, less accurate, and accurate and efficient, respectively. We found that OTC-GRPO and Search-R1-PPO achieves same results with same tool calls in 42.54% cases of HotpotQA on Qwen2.5-3B-Base model, leading to relatively lower ME.

| Method | NQ | | | | | HotpotQA | | | | |
|---|---|---|---|---|---|---|---|---|---|---|
| | ME ($\uparrow$) | LE ($\downarrow$) | MA ($\uparrow$) | LA ($\downarrow$) | AE ($\uparrow$) | ME ($\uparrow$) | LE ($\downarrow$) | MA ($\uparrow$) | LA ($\downarrow$) | AE ($\uparrow$) |
| **Qwen2.5-3B(-Base)** | | | | | | | | | | |
| OTC-PPO | 63.55 | 0.02 | 4.60 | 7.67 | 3.74 | 61.78 | 0.53 | 4.30 | 6.34 | 3.35 |
| OTC-GRPO | 65.43 | 2.49 | 6.15 | 3.85 | 5.10 | 35.54 | 5.96 | 11.89 | 4.07 | 3.47 |
| **Qwen2.5-7B(-Base)** | | | | | | | | | | |
| OTC-PPO | 86.2 | 0.03 | 6.45 | 6.92 | 6.45 | 81.49 | 0.08 | 8.60 | 8.68 | 8.53 |
| OTC-GRPO | 82.0 | 0.00 | 7.26 | 10.69 | 7.26 | 79.71 | 0.00 | 8.15 | 11.98 | 8.12 |

of TC is also various across the datasets, which relates to many factors such as inherent capabilities of different models (i.e., self-awareness) and complexity of different datasets, leading to varying minimal number of tool calls for each question and model; iv) As the model size increases, the TC and TP get bigger boost no matter in OTC-PPO or OTC-GRPO. We highlight the ii) and iv) are very important since they compose the great potential scalability for our methods as model scales.

**Code as Tool.** We mainly follow the same setting and only report the GRPO results to be consistent with original ToRL (Li et al., 2025b) paper. Table 2 shows the results. We observe several similar findings: i) our method does not sacrifice the accuracy too much and even brings some improvement when the model gets larger. We attribute this to the development of internal reasoning capability of models when it is enforced to minimizing external tool calls, as evidenced in our case study; ii) our method can significantly boost the tool productivity, reducing the unnecessary tool calls; iii) As model size increases, our method is more effective to improve the tool productivity.

## 5 ANALYSIS

We mainly conduct our analysis using search as a tool in this section and leave more analysis in the Appendix F and G, respectively.

### 5.1 TOOL USE BEHAVIOR ANALYSIS

**Training.** Fig. 2 shows the training behaviors of different methods. It is clear that our method not only achieves comparable results with fewer tool calls and shorter responses, but also enables faster and more efficient training optimization. This is particularly important, as it significantly reduces the time and cost associated with real-time tool interactions during training, both in terms of financial expenses and computational overhead. We also find GRPO tends to achieve better performance than PPO due to more accurate approximation of minimal tool call required, but it is less stable than PPO, as also evidenced by several recent studies (Jin et al., 2025). It is encouraging to find that our method can delay the early collapse substantially with more training steps.

**Inference.** We identify several representative tool-use behavior types of our proposed method: i) the answer is same with *less* tool calls compared with baseline (*more efficient*); ii) the answer is same with *more* tool calls compared with baseline (*less efficient*); and iii) our method is able to produce the correct answer whereas the baseline fails to answer correctly (*more accurate*); iv) the baseline

Table 4: The results of Out-of-Domain (OOD) evaluation of OTC against Search-R1 in EM and TC.

| Models | TriviaQA | | PopQA | | 2Wiki | | Musique | | Bamboogle | |
|---|---|---|---|---|---|---|---|---|---|---|
| | EM (↑) | TC (↓) | EM (↑) | TC (↓) | EM (↑) | TC (↓) | EM (↑) | TC (↓) | EM (↑) | TC (↓) |
| **Qwen2.5-3B(-Base)** | | | | | | | | | | |
| Search-R1-PPO | 0.566 | 1.580 | 0.425 | 1.631 | 0.258 | 1.675 | 0.051 | 1.922 | 0.063 | 1.766 |
| Search-R1-GRPO | 0.587 | 1.455 | 0.345 | 1.542 | 0.257 | 1.991 | 0.084 | 2.263 | 0.203 | 1.859 |
| OTC-PPO | 0.551 | **1.008** | 0.409 | **1.009** | 0.235 | **1.050** | 0.045 | 1.051 | 0.063 | **1.016** |
| OTC-GRPO | **0.608** | 1.046 | **0.441** | 1.030 | **0.341** | 1.561 | **0.124** | 1.734 | **0.266** | 1.547 |
| **Qwen2.5-7B(-Base)** | | | | | | | | | | |
| Search-R1-PPO | 0.596 | 3.353 | 0.420 | 3.315 | 0.326 | 4.116 | 0.135 | 4.294 | 0.375 | 3.641 |
| Search-R1-GRPO | 0.578 | 1.704 | 0.411 | 1.754 | 0.340 | 2.521 | 0.130 | 2.616 | 0.203 | 1.859 |
| OTC-PPO | **0.623** | 1.066 | 0.425 | 1.083 | **0.363** | 1.868 | **0.152** | 1.942 | **0.391** | 1.828 |
| OTC-GRPO | 0.597 | **0.430** | **0.431** | **0.739** | 0.311 | **0.938** | 0.130 | **1.224** | 0.250 | **0.781** |

Table 5: The results of OTC-PO under Qwen2.5-7B-Base and Qwen2.5-7B-Instruct models.

| Models | NQ | | | HotpotQA | | |
|---|---|---|---|---|---|---|
| | EM (↑) | TC (↓) | TP (↑) | EM (↑) | TC (↓) | TP (↑) |
| **Qwen2.5-7B(-Base)** | | | | | | |
| OTC-PPO | **0.446** | 1.040 | 0.429 | 0.383 | 1.464 | 0.262 |
| OTC-GRPO | 0.444 | **0.990** | **0.448** | 0.366 | **1.005** | **0.364** |
| **Qwen2.5-7B(-Instruct)** | | | | | | |
| OTC-PPO | 0.389 | 1.404 | 0.277 | 0.381 | 1.880 | 0.203 |
| OTC-GRPO | 0.429 | 1.322 | 0.325 | **0.386** | 1.956 | 0.197 |

can produce the correct answer whereas the our method fails to answer correctly (*less accurate*); and v) our method is able to produce the correct answer using fewer tool calls whereas the baseline fails to answer correctly (*accurate and efficient*). Table 3 shows the results compared with the baseline Search-R1-PPO [3]. On the efficiency side, it is clear to find that our method achieve same answers with baseline using fewer tool calls in most cases, and more than 80% when the model becomes larger. On the effectiveness side, we conclude that there is no significant loss when using same RL algorithm, as evidenced by lower gap between MA and LA in OTC-PPO method. Moreover, among all MA cases, it is observed that AE accounts for roughly 80% in 3B model (i.e., 3.74/4.60 =81%), and exceeds 95% on 7B model. These results highlight the great strength of OTC-PO in promoting efficiency while reserving effectiveness.

## 5.2 Out-of-domain Evaluation

We run out-of-domain evaluation on more QA benchmarks. Fig. 6 shows the results of TP and Table 4 shows the results of EM and TC. Generally, we find that our method achieves highest TP across all the model and benchmark. More importantly, we observe that OTC-GRPO is better than OTC-PPO on TP and as the model size increases, our method showcases bigger improvement, serving as another evidence of great scalability of our method. Notably, we find that OTC-PPO outperforms the Search-R1-PPO in terms of EM on all benchmarks with Qwen2.5-7B-Base model with much less tool calls, demonstrating the better generalization capabilities of our methods.

## 5.3 More Analysis

**Base v.s. Instruct Model.** We additionally compare the performance of OTC-PO using Qwen-7B-Base and Qwen-7B-Instruct models. Table 5 shows the final results. It is observed that i) Base models mostly lead to better performance, which is also consistent with results in the Search-R1 paper; ii) Instruct models tends to use more tools compared with base model while the TC still is much lower than the Search-R1. We attribute this to the Instruct model tends to memorize specific pattern during supervised fine-tuning and easily to be triggered to use tools even unnecessary.

**Case Study** Fig. 3 presents a case study to better illustrate the advantages of our proposed OTC-PO. The analysis highlights several observations: (i) Search-R1 tends to issue multiple redundant search calls, reflecting the absence of an explicit incentive to limit tool usage; (ii) both OTC-PPO and OTC-GRPO substantially reduce unnecessary tool calls, but they do so in different ways. OTC-PPO improves efficiency by generating more informative queries, while OTC-GRPO relies more on

---

[3]We use the same baseline to better distinguish OTC-PPO and OTC-GRPO.

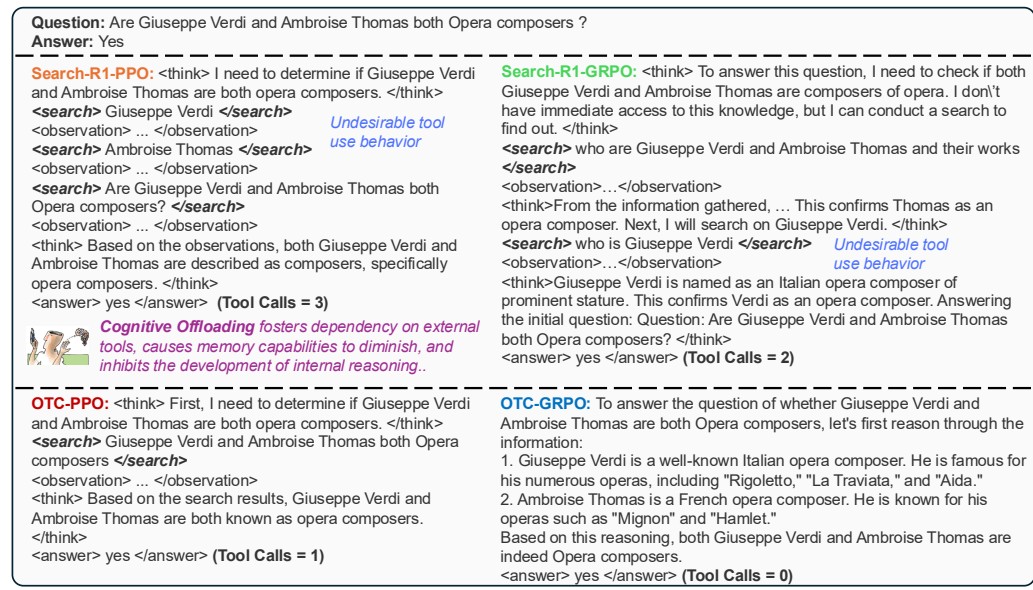

**Question:** Are Giuseppe Verdi and Ambroise Thomas both Opera composers ?
**Answer:** Yes

**Search-R1-PPO:** <think> I need to determine if Giuseppe Verdi and Ambroise Thomas are both opera composers. </think>
*<search>* Giuseppe Verdi *</search>*
<observation> ... </observation> *Undesirable tool use behavior*
*<search>* Ambroise Thomas *</search>*
<observation> ... </observation>
*<search>* Are Giuseppe Verdi and Ambroise Thomas both Opera composers? *</search>*
<observation> ... </observation>
<think> Based on the observations, both Giuseppe Verdi and Ambroise Thomas are described as composers, specifically opera composers. </think>
<answer> yes </answer> **(Tool Calls = 3)**

*Cognitive Offloading fosters dependency on external tools, causes memory capabilities to diminish, and inhibits the development of internal reasoning..*

**Search-R1-GRPO:** <think> To answer this question, I need to check if both Giuseppe Verdi and Ambroise Thomas are composers of opera. I don\'t have immediate access to this knowledge, but I can conduct a search to find out. </think>
*<search>* who are Giuseppe Verdi and Ambroise Thomas and their works *</search>*
<observation>…</observation>
<think>From the information gathered, … This confirms Thomas as an opera composer. Next, I will search on Giuseppe Verdi. </think>
*<search>* who is Giuseppe Verdi *</search>* *Undesirable tool use behavior*
<observation>…</observation>
<think>Giuseppe Verdi is named as an Italian opera composer of prominent stature. This confirms Verdi as an opera composer. Answering the initial question: Question: Are Giuseppe Verdi and Ambroise Thomas both Opera composers? </think>
<answer> yes </answer> **(Tool Calls = 2)**

**OTC-PPO:** <think> First, I need to determine if Giuseppe Verdi and Ambroise Thomas are both opera composers. </think>
*<search>* Giuseppe Verdi and Ambroise Thomas both Opera composers *</search>*
<observation> ... </observation>
<think> Based on the search results, Giuseppe Verdi and Ambroise Thomas are both known as opera composers. </think>
<answer> yes </answer> **(Tool Calls = 1)**

**OTC-GRPO:** To answer the question of whether Giuseppe Verdi and Ambroise Thomas are both Opera composers, let's first reason through the information:
1. Giuseppe Verdi is a well-known Italian opera composer. He is famous for his numerous operas, including "Rigoletto," "La Traviata," and "Aida."
2. Ambroise Thomas is a French opera composer. He is known for his operas such as "Mignon" and "Hamlet."
Based on this reasoning, both Giuseppe Verdi and Ambroise Thomas are indeed Opera composers.
<answer> yes </answer> **(Tool Calls = 0)**

Figure 3: A case study using *search* as a tool on Qwen-7B-Base model, where all four methods arrive at the correct answer but exhibit distinct behaviors in internal reasoning and external tool usage. OTC-GRPO does not rely on external tools while Search-R1-PPO uses extensive tool calls.

internal reasoning when the problem can be solved without external tools. Overall, the case study suggests that without guidance, models may overuse external tools, whereas encouraging minimal tool calls while preserving correctness promotes more balanced and deliberate tool use.

**Reasoning v.s. Acting.** We further examine the relationship between reasoning and tool use when models are encouraged to reduce external calls. Specifically, we compare OTC-PPO with Search-R1-PPO by recording, for each correctly answered sample, both the number of tool calls and the number of reasoning tokens (excluding search and observation tokens in response). As shown in Figure 4, OTC-PPO generally makes fewer tool calls while producing longer reasoning traces. Moreover, this difference becomes more pronounced as model size increases, suggesting that larger models are better able to leverage internal reasoning for external tool use under our framework [4].

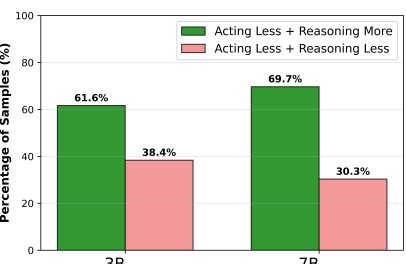

Figure 4: Comparison of reasoning (number of reasoning tokens) and acting (number of tool calls) between Search-R1-PPO and OTC-PPO on the NQ.

## 6    CONCLUSION AND FUTURE WORK

In this work, we redefine the objective of task for agentic RL not only provide the final correct answer, but also optimize the tool use behavior of model to achieve such goal. We then introduce OTC-PO, a simple yet effective RL framework that explicitly encourages LLMs to generate correct answers with fewer tool calls. Unlike prior work that primarily focuses on final answer correctness, our approach incorporates a tool-integrated reward that accounts for both effectiveness and efficiency of tool usage, thereby promoting tool productivity without sacrificing the accuracy a lot in several benchmarks with different tools. We also find that extensively rely on external tools hinder the development and utilization of internal reasoning capabilities of the model, and minimizing external tool calls alternatively foster the development of internal reasoning capabilities. Finally, in future work, we aim to extend our framework to more complex agentic tasks involving a broader set of tools and longer-horizon reasoning.

---

[4]More case studies, hyper-parameter analysis can be found in Appendix.

## REPRODUCIBILITY STATEMENT

We strictly follow the same setting of two popular open-sourced codebases: Search-R1 (`https://github.com/PeterGriffinJin/Search-R1`) and ToRL (`https://github.com/GAIR-NLP/ToRL`), and use same hyper-parameters and data to conduct our main experiments. We attach our reward design code in the supplementary for reproduction. The details for reproducing results are also mentioned in Sections 4.1 of the main paper.

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

## A  THE USE OF LARGE LANGUAGE MODELS (LLMs)

LLMs did not play significant roles in this paper's research ideation and/or writing to the extent that they could be regarded as a contributor. In the experiments, LLMs are treated as the main experimental object.

## B  RELATED WORK

**Reward Shaping in RL.**  Reward shaping plays a critical role in reinforcement learning, as it directly shapes the behavior the model learns to optimize (Laud, 2004; Wang et al., 2022). Recent advances have introduced several reward signals for LLMs to consider the correctness and the internal knowledge-only reasoning efficiency together such as the length of responses and difficulty of the questions (Aggarwal & Welleck, 2025; Arora & Zanette, 2025; Wang et al., 2025b). For instances, Length Controlled Policy Optimization (LCPO) (Aggarwal & Welleck, 2025) is proposed to satisfy the length constraints while optimizing reasoning performance and some efforts try to dynamically allocate inference time compute based on task complexity (Arora & Zanette, 2025; Hou et al., 2025).

## C  TOOL-INTEGRATED REINFORCEMENT LEARNING

**PPO in TRL.**  Proximal Policy Optimization (PPO) is a widely used policy-gradient method in lots of tasks (Ouyang et al., 2022). Given our formulation, PPO updates the policy, using trajectories sampled from the previous policy, and maximizes the following objective:

$$\mathcal{J}_{\text{PPO}}(\theta) = \mathbb{E}_{q \sim \mathcal{D}, \, y \sim \pi_{\text{old}}} \left[ \frac{1}{\sum_{t=1}^{|y|} \mathbb{I}(y_t)} \sum_{t=1}^{|y|} \mathbb{I}(y_t) \cdot \min \left( \rho_t A_t, \, \text{clip}(\rho_t, 1 - \epsilon, 1 + \epsilon) A_t \right) \right], \quad (8)$$

where $\pi_\theta$ and $\pi_{old}$ are current and previous policy models, and $p_t = \frac{\pi_\theta(y_t|q, y_{<t}; \mathcal{E})}{\pi_{old}(y_t|q, y_{<t}; \mathcal{E})}$. Here, $\mathbb{I}(y_t)$ is an indicator function marking whether token $y_t$ is generated by the model (i.e., $r_i$ and $tc_i$) or returned from the environment $\mathcal{E}$ (i.e., $o_i$). The advantage estimate $A_t$ is computed via Generalized Advantage Estimation (GAE) and $\epsilon$ a PPO clipping threshold to constrain the policy update.

**GRPO in TRL.**  To improve the stability of policy optimization in language models and avoid reliance on an additional value function approximation, Group Relative Policy Optimization (GRPO) is introduced which uses the relative ranking of multiple sampled outputs as a baseline for computing advantages, rather than fitting a value function. For each input question $q$, GRPO samples a group of $G$ response $\{y_1, y_2, ..., y_G\}$ from the reference policy $\pi_{ref}$. The trained policy $\pi_\theta$ is then updated by maximizing the following objective:

$$\mathcal{J}_{\text{GRPO}}(\theta) = \mathbb{E}_{q \sim \mathcal{D}, \, \{y_i\}_{i=1}^G \sim \tau_{\text{old}}(\cdot|q; \mathcal{E})} \left[ \frac{1}{G} \sum_{i=1}^{G} \frac{1}{\sum_{t=1}^{|y_i|} \mathbb{I}(y_{i,t})} \sum_{t=1}^{|y_i|} \mathbb{I}(y_{i,t}) \right.$$
$$\left. \cdot \min \left( p_t \hat{A}_{i,t}, \, \text{clip}\left( p_t, 1 - \epsilon, \, 1 + \epsilon \right) \hat{A}_{i,t} \right) \right] - \beta \, \mathbb{D}_{\text{KL}}[\pi_\theta \, \| \, \pi_{\text{ref}}], \quad (9)$$

Here, $p_t = \frac{\pi_\theta(y_{i,t}|x, y_{i,<t}; \mathcal{R})}{\pi_{\text{old}}(y_{i,t}|x, y_{i,<t}; \mathcal{R})}$, $\hat{A}_{i,t}$ denotes the advantage at token $t$ in response $y_i$, computed based on the relative ranking of rewards within the group, $\beta$ controlling the KL-regularization strength. The clipping threshold $\epsilon$ ensures stable updates.

## D    SEARCH TEMPLATE

Answer the given question. You must conduct reasoning inside <think> and </think> first every time you get new information. After reasoning, if you find you lack some knowledge, you can call a search engine tool by <search> query </search>, and it will return the top searched results between <observation> and </observation>. You need to make every search call count and gain helpful results. If you find no further external knowledge needed, you can directly provide the answer inside <answer> and </answer> without detailed illustrations. For example, <answer> xxx </answer>. Question: question.

## E    REWARD FUNCTION

### E.1    FIGURE ILLUSTRATION

We draw the two types of $r_{tool}$ defined in the main content for better understanding. Fig. 5 shows the illustration of these reward functions for OTC-PPO and OTC-GRPO respectively. It is very clear that: 1) Left: as the number of tool calls increases, the $r_{tool}$ decreases accordingly. Thus when multiple trajectories leads to correct answer, the one that use less tools will get higher reward and the one that do not use tool will get the highest reward; 2) Right: It is obvious that the diagonal achieves the highest reward as the number of tool calls is the less one, and as the $m$ increases, the color brightness becomes larger since multiple tool calls are involved.

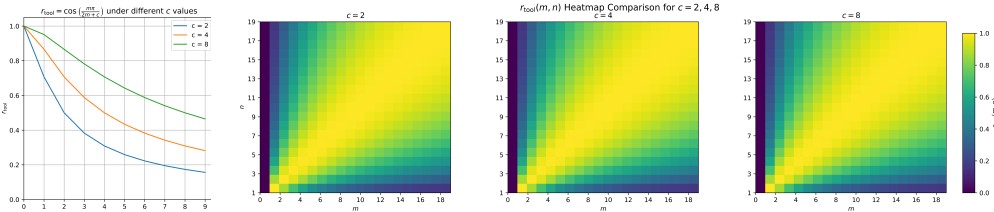

Figure 5: **Left**: the illustration of tool-use reward functions in OTC-PPO; and **Right**: the illustration of tool-use reward functions in OTC-GRPO.

### E.2    THEORETICAL JUSTIFICATION

We mainly follow the justification (Section 4.2) in the paper (Arora & Zanette, 2025) to showcase our proposed reward design leads to reducing the tool calls without compromising accuracy. We strongly encourage readers to refer to the previous paper for complete details. All assumptions made in that work still hold in our setting, with the only difference being that our response is defined as $y = (\tau, a)$, which includes both the tool-integrated reasoning trajectory and the final answer, rather than language-only reasoning.

Let $\theta^*_{eff}$ denote the population-level parameters of the policy models obtained by maximizing Equation 7, i.e.,

$$\theta^\star_{\text{eff}} = \arg\max_\theta \left\{ \mathbb{E}_{x \sim p} \mathbb{E}_{y \sim p_\theta(x)} \left[ (\alpha * r_{tool}) \right] \right\} \tag{10}$$

as $r_\phi(q, y)$ is mainly the indicator function about the correctness and format. Therefore, the population-level maximizer $p_{\theta^*_{eff}}$ is as accurate as the population-level maximizer $p^*_\theta$ and Acc $(p^*_{\theta_{eff}}) = 1$.

### E.3    GENERALIZATION OF REWARD

We use two major forms of $r_\phi$: i) $r^1_\phi = r_{correct}$; and ii) $r^2_\phi = r_{correct} + r_{format}$ to illustrate the generalization of our proposed tool-integrated reward design $r^{tool}_\phi = \alpha * r_{tool} * r_\phi$.

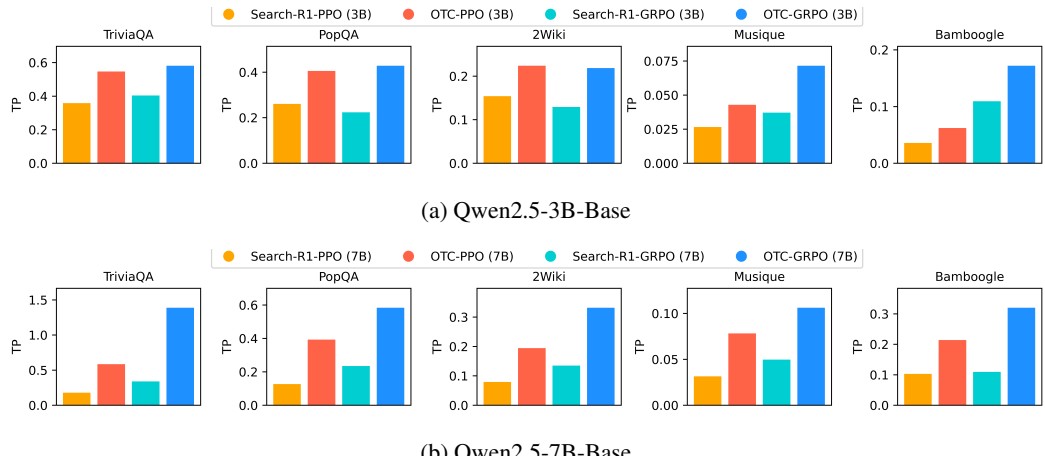

Figure 6: The Out-of-domain performance of OTC-PO and Search-R1 in TP.

Table 6: The results of OTC-PO with different $\alpha$ on Qwen2.5-7B-Base model

| Models | NQ | | | HotpotQA | | |
|---|---|---|---|---|---|---|
| | EM ($\uparrow$) | TC ($\downarrow$) | TP ($\uparrow$) | EM ($\uparrow$) | TC ($\downarrow$) | TP ($\uparrow$) |
| $\alpha = 1$ | 0.446 | 1.040 | 0.429 | 0.383 | 1.464 | 0.262 |
| $\alpha = 2$ | 0.354 | 1.571 | 0.225 | 0.320 | 2.062 | 0.155 |
| $\alpha = 3$ | 0.389 | 1.530 | 0.254 | 0.340 | 1.948 | 0.175 |

*If the answer is correct*, the $r_\phi^1$ and $r_\phi^2$ are both positive and fixed for all questions, therefore the $r_\phi^{tool}$ is only determined by the number of $\alpha$ and $r_{tool}$. Thus it can assign higher score for less tools and lower score for more tools by definition of $r_{tool}$.

*If the answer is wrong*, the $r_\phi^1$ will be 0, and therefore disables the influence of $r_{tool}$, reducing the reward hacking issue. The $r_\phi^2$ will be $r_{format}$ and then $r_\phi^{tool}$ becomes $\alpha * r_{tool} * r_{format}$ which is acceptable. Considering two cases of $r_{format}$, if $r_{format}$ is 0, then the final reward is also 0; and if $r_{format}$ is a positive, the $r_\phi^{tool}$ still holds as less tools will be assigned more rewards. This is reasonable since less tool calls means less cost especially when the answer is wrong. We note that the reward gap here between wrong answer (i.e., $r_{format}$) and correct answer ($i.e., r_{correct} + r_{format}$) is significant for the policy model to learn the desirable behavior.

# F  ANALYSIS OF SEARCH AS TOOL

## F.1  THE EFFECTS OF $\alpha$

Table 6 shows the performance of OTC-PO with different $\alpha$. It can be found that increasing $\alpha$ can not leads to better results. We further check the reward changes during the training and find that it becomes harder for the model to learn the desired behavior when using a larger $\alpha$, because a single tool call can cause large fluctuations in the reward, especially considering the relatively lower $C$. We suspect the $\alpha$ will be more important in much longer tool-integrated reasoning trajectories.

# G  ANALYSIS OF CODE AS TOOL

## G.1  THE EFFECTS OF $C$

Table 7 shows the effects of $C$ in ToRL. It is observed that i) the larger $C$ always leads to more tool calls on ToRL-GRPO if we only consider the final correctness as the reward and do not penalize the tool use behaviors of LLMs; ii) OTC-GRPO achieves more stable tool calls which is more reasonable as the optimal number of tool calls should not be affected by $C$, and leads to bigger TP

Table 7: The effects of $C$ on Qwen2.5-Math-7B Base model.

| Models | AIME24 | | | AIME25 | | |
|---|---|---|---|---|---|---|
| | EM ($\uparrow$) | TC ($\downarrow$) | TP ($\uparrow$) | EM ($\uparrow$) | TC ($\downarrow$) | TP ($\uparrow$) |
| *C = 1* | | | | | | |
| ToRL-GRPO | 30.0 | 0.9 | 33.3 | 26.7 | 0.9 | 29.7 |
| OTC-GRPO | 35.8 | 0.8 | 44.8 (▲ 34.5%) | 26.7 | 0.8 | 33.4 (▲ 11.1%) |
| *C = 2* | | | | | | |
| ToRL-GRPO | 33.3 | 1.4 | 23.8 | 25.8 | 1.4 | 18.4 |
| OTC-GRPO | 33.3 | 1.0 | 33.3 (▲ 39.9%) | 23.3 | 0.7 | 33.3 (▲ 81.0%) |
| *C = 3* | | | | | | |
| ToRL-GRPO | 36.7 | 2.1 | 17.5 | 26.7 | 2.1 | 12.7 |
| OTC-GRPO | 36.7 | 0.7 | 52.4 (▲ 199.4%) | 23.3 | 0.8 | 29.1 (▲ 129.1%) |

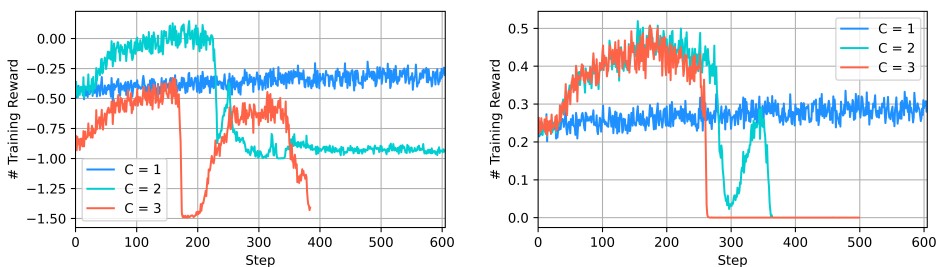

Figure 7: **Left**: Training reward changes of ToRL-GRPO; and **Right**: Training reward changes of OTC-GRPO.

Table 8: The results of OTC-PO on MATH when $C$ = 3.

| Models | MATH | | |
|---|---|---|---|
| | EM ($\uparrow$) | TC ($\downarrow$) | TP ($\uparrow$) |
| **Qwen2.5-Math-1.5B(-Base)** | | | |
| ToRL-GRPO | 52.2 | 1.8 | 29.0 |
| OTC-GRPO | 53.7 | 1.1 (▼ 38.9%) | 48.8 (▲ 68.28%) |
| **Qwen2.5-Math-7B(-Base)** | | | |
| ToRL-GRPO | 59.7 | 1.49 | 40.07 |
| OTC-GRPO | 61.9 | 0.68 (▼ 54.4%) | 91.03 (▲ 127.2%) |

improvement as $C$ increases. Further, Fig. 7 shows the rewards changes with different $C$. It can be found that GRPO tends to collapse earlier with bigger $C$ and our OTC-PO can delay this, as also observed using search as tool.

## G.2 Additional Results

We additionally run the evaluation on MATH dataset following ToRL (Li et al., 2025b), and Table 8 shows the final results. The conclusion also holds as analyzed in the main experiments, revealing the great generalization capability of our method.

# H CASE STUDY

## H.1 MORE EXAMPLES

Fig. 8 and Fig. 9 show more examples between Search-R1 and OTC-PO methods. Fig. 10 and Fig. 11 show more examples between ToRL-GRPO and OTC-GRPO. It is clear that OTC-PO can leverage more reasoning capabilities while minimizing external tool calls.

---

**Case Study: No Tool Call Required**

Question: The Bellagio resort owned by MGM Resorts international is located in which US city?
Answer: Las Vegas

**Search-R1-PPO**: <think> I need to find the US city where the Bellagio resort owned by MGM Resorts International is located. </think>
<search> Bellagio resort owned by MGM Resorts International </search>
<observation> . . . </observation>
<search> Bellagio resort city </search>
<observation> . . . </observation>
<think> Based on the observations, the Bellagio resort owned by MGM Resorts International is located in Las Vegas, Nevada. </think>
<answer> Las Vegas </answer> **(Tool Calls = 2)**

**OTC-GRPO**: To answer this question, let's first reason through the information provided:
1. We need to find the US city where the Bellagio resort, owned by MGM Resorts International, is located.
2. The Bellagio resort is a well-known hotel and casino located in Las Vegas, Nevada.
3. Therefore, the US city where the Bellagio resort is located is Las Vegas.
<answer> Las Vegas </answer> **(Tool Calls = 0)**

---

Figure 8: An example that OTC-GRPO provide correct answer solely rely on its own internal reasoning without any external tool calls while Search-R1 extensively call unnecessary tools.

---

**Case Study: Less Tool Calls with Accurate Search**

Question: What science fantasy young adult series, told in first person, has a set of companion books narrating the stories of enslaved worlds and alien species?
Answer: Animorphs

**Search-R1-PPO**: <think> I need to find a science fantasy young adult series that is told in first person and has a set of companion books narrating the stories of enslaved worlds and alien species. </think>
<search> science fantasy young adult series told in first person </search>
<observation> ... <observation>
<search> companion books animorphs enslaved worlds alien species </search>
<observation> ... </observation>
<search> science fantasy young adult series told in first person enslaved worlds alien species companion books </search>
<observation> ... </observation>
<think> Based on the observations, the science fantasy young adult series told in first person and has a set of companion books narrating the stories of enslaved worlds and alien species is "Animorphs". </think>
<answer> Animorphs </answer> **(Tool Calls = 3)**

**OTC-GRPO**: To answer this question, let's first reason through the information provided:
1. We need to find a science fantasy young adult series.
2. The series is told in first person.
3. It has a set of companion books narrating the stories of enslaved worlds and alien species.
Let's search for this information:
<search> What science fantasy young adult series, told in first person, has a set of companion books narrating the stories of enslaved worlds and alien species? </search>
<observation> ... </observation>
Based on the search results, the science fantasy young adult series that fits the description is "Animorphs".
<answer> Animorphs </answer> **(Tool Calls = 1)**

Figure 9: An example that OTC-GRPO uses less tool calls but with more accurate search to arrive a correct answer.

**Question:** Every morning Aya goes for a $9$-kilometer-long walk and stops at a coffee shop afterwards. When she walks at a constant speed of $s$ kilometers per hour, the walk takes her 4 hours, including $t$ minutes spent in the coffee shop. When she walks $s+2$ kilometers per hour, the walk takes her 2 hours and 24 minutes, including $t$ minutes spent in the coffee shop. Suppose Aya walks at $s+\frac{1}{2}$ kilometers per hour. Find the number of minutes the walk takes her, including the $t$ minutes spent in the coffee shop. **Answer:** 204

**ToRL-GRPO:** To solve this problem, we need to determine …

```python
from sympy import symbols, Eq, solve
s = symbols('s')
equation = Eq(4 - 9/s, 2.4 - 9/(s+2))
s_solution = solve(equation, s)
print(s_solution)
```
```output
[-4.50000000000000, 2.50000000000000]
```

The solutions for \(s\) are \(-4.5\) and \(2.5\) … Let's confirm the solution…

```python
...
time_spent_walking = 9 / new_speed
total_time_minutes = (time_spent_walking * 60) + t_value
print(total_time_minutes)
```
```output
204.0
```

The total time it takes for Aya to walk at a speed of \(3\) km/h, including the \(24\) minutes spent in the coffee shop, is \(\boxed{204}\) **(Tool Calls = 2)**

**OTC-GRPO:** To solve this problem, we need to determine …

```python
import math
a = 1.6, b = 3.2, c = -18
discriminant = b**2 - 4*a*c
s1 = (-b + math.sqrt(discriminant)) / (2*a)
s2 = (-b - math.sqrt(discriminant)) / (2*a)
print((s1, s2))
```
```output
(2.4999999999999996, -4.499999999999999)
```

Since speed cannot be negative, we have \(s = 2.5\) km/h.\n\nNow we can substitute \(s = 2.5\) back into one of the original equations to find \(t\)). Using the first equation:\n\[\n\frac{9}{2.5} + \frac{t}{60} = 4\n\]\n\[\n3.6 + \frac{t}{60} = 4\n\]\n\[\n\frac{t}{60} = 0.4\n\]\n\[\nt = 0.4 \times 60 = 24\n\]\n\nSo, \(t = 24\) minutes. Now we need to find the total time when Aya walks at \(s + \frac{1}{2} = 2.5 + 0.5 = 3\) km/h. The time spent walking is \(\frac{9}{3} = 3\) hours, and the total time including the time spent in the coffee shop is \(3 + \frac{24}{60} = 3 + 0.4 = 3.4\) hours, which is 3 hours and 24 minutes.\n\nThus, the number of minutes the walk takes her, including the \(t\) minutes spent in the coffee shop, is \(3 \times 60 + 24 = 204\) minutes. The final answer is: \(\boxed{204}\)

**(Tool Calls = 1)**

Figure 10: An example of OTC-GRPO use *less tool calls* to get the correct answer.

**Question:** Find all values of $x$ that satisfy the equation $x = \!\!\sqrt{11-2x} + 4$.
**Answer:** 5

**ToRL-GRPO:** To solve the equation \(x = \sqrt{11 - 2x} + 4\), we can follow these steps ……

```python
import sympy as sp
x = sp.symbols('x')
equation = x - (sp.sqrt(11 - 2*x) + 4)
isolated_sqrt = sp.Eq(x - 4, sp.sqrt(11 - 2*x))
squared_equation = sp.Eq((x - 4)**2, 11 - 2*x)
simplified_equation = sp.simplify(squared_equation)
solutions = sp.solve(simplified_equation, x)
valid_solutions = [sol for sol in solutions if equation.subs(x, sol) == 0]
print(valid_solutions)
```
```output
[5]
```

The solution to the equation \(x = \sqrt{11 - 2x} + 4\) is \(x = 5\). Therefore, the final answer is \(\boxed{5}\).

**(Tool Calls = 1)**

**OTC-GRPO:** To solve the equation \(x = \sqrt{11 - 2x} + 4\), we can follow these steps:

1. Isolate the square root term.
2. Square both sides to eliminate the square root.
3. Solve the resulting equation.
4. Verify the solutions by substituting back into the original equation to ensure they satisfy it.

Let's go through these steps systematically.

1. Isolate the square root term:\n\[ x - 4 = \sqrt{11 - 2x} \]
2. Square both sides:\n\[ (x - 4)^2 = 11 - 2x \]
3. Expand and simplify the equation:\n\[ x^2 - 8x + 16 = 11 - 2x \]\n\[ x^2 - 6x + 5 = 0 \]
4. Solve the quadratic equation:\n\[ (x - 5)(x - 1) = 0 \]

So, the solutions are \(x = 5\) and \(x = 1\).

5. Verify the solutions by substituting back into the original equation.
For \(x = 5\):\n\[ 5 = \sqrt{11 - 2 \cdot 5} + 4 \]\n\[ 5 = \sqrt{1} + 4 \]\n\[ 5 = 1 + 4 \]\n\[ 5 = 5 \]\nThis is true, so \(x = 5\) is a valid solution.\n\nFor \(x = 1\):\n\[ 1 = \sqrt{11 - 2 \cdot 1} + 4 \]\n\[ 1 = \sqrt{9} + 4 \]\n\[ 1 = 3 + 4 \]\n\[ 1 = 7 \]\nThis is false, so \(x = 1\) is not a valid solution.\n\nTherefore, the only valid solution is \(x = 5\).\n\nThe final answer is: \(\boxed{5}\)

**(Tool Calls = 0)**

Figure 11: An example of OTC-GRPO *does not need any external tool calls* to get the correct answer.