# OpenReview forum: "Teaching Language Model to Act Efficiently"
_ICLR.cc/2026/Conference — Submitted to ICLR 2026_

### Official Review · Reviewer_oxjH · 2025-10-28

**Soundness:** 4
**Presentation:** 3
**Contribution:** 3
**Rating:** 6
**Confidence:** 5

**Summary:**

The authors investigate tool-integrated reasoning large language models, and introduce the Optimized Tool Call-controlled Policy Optimization (OTC-PO) method to handle the cognitive offloading issue. The core to OTC-PO is a novel reward function that relates to tool calls. By using the proposed reward function, OTC-PO  not only improves the performances but also reduce the number of tool calls.

**Strengths:**

- The paper is well written and easy to understand.
- The authors provide sufficient motivation for the proposed reward function.
- Extensive experiments are conducted to validate the effectiveness of **OTC-PO**.

**Weaknesses:**

- The reviewer would like to know the practical cost of tool invocation. According to **Table 1**, **Search-R1-PPO** calls the tool up to three times, while **OTC-PPO** also requires at least one call. What is the actual time savings achieved in this case? This investigation is essential to assess the real-world significance of the proposed method.
- The reviewer suggests conducting experiments on larger-scale models, if computational resources permit, to further verify the scalability of the approach.
- An important related work on **tool-augmented LLMs** [1] is missing from the discussion in the paper.

[1] Advancing Tool-Augmented Large Language Models: Integrating Insights from Errors in Inference Trees. NeurIPS 2024.

**Questions:**

See Weaknesses.

---

> ### Author Response · Authors · 2025-11-24
>
> Thank you for your encouraging review and specific suggestions. To fully address your concern, we provide detailed response as follows:
>
> Weakness:
>
> > W1: time analysis
>
> - We provide running time analysis during training at Figure 2 (middle). In summary, when trained all methods for 400 steps, OTC-PPO reduces training time by roughly 10% compared to Search-R1-PPO, while OTC-GRPO reduces runtime by roughly 25% relative to Search-R1-GRPO. It is noted that fewer tool calls not only save computation time but also reduce monetary cost for systems that rely on paid API services. Even at inference, where the number of samples is much smaller than in training, our method still yields about a 10% time reduction. Since the cost is highly sensitive to the chosen service (i.e., local server or API services), we alternatively report the average number of tool call for better comparison.
>
> >W2: experiments on bigger model
>
> - Thank you for the suggestion. We additionally run our method at 14B model while 32B is too large to run considering our limited resources and tight time windows. The results can be found in follows:
>
>     | Methods | NQ-EM(&uarr;) | NE-TC(&darr;) | NQ-TP(&uarr;) | HotpotQA-EM(&uarr;) | HotpotQA-TC(&darr;) | HotpotQA-TP(&uarr;) |
>     |------|------|------|------|------|------|------|
>     |  Search-R1-PPO    |  0.450   | 3.200    |   0.141   |  0.408    |  3.010   |   0.136   |
>     |  OTC-PPO   |  0.432    |  1.680    |  0.257 (&uarr;82.3%)    |  0.410    | 1.950     |   0.210 (&uarr;54.5%)   |
>
>     It is evident that our method can scale large models and improve tool productivity.
>
> >W3: missing references
>
> - We already add the recommended paper in the discussion (line 100-101). Thank you for your recommendation.

---

### Official Review · Reviewer_y9uq · 2025-10-28

**Soundness:** 3
**Presentation:** 3
**Contribution:** 3
**Rating:** 6
**Confidence:** 3

**Summary:**

The paper introduces OTC-PO, an RL framework that trains tool-using LLM agents to optimize both answer correctness and efficiency of tool use. The key idea proposed by the authors is a tool-integrated reward that multiplies a correctness term with a tool-efficiency coefficient that favors trajectories reaching correct answers with fewer tool calls.

Tool-integrated agents increasingly dominate practical workflows. Optimizing only EM produces agents that over-call tools, inflate latency, and offload trivial reasoning to external systems. This paper reframes training as a compute-aware objective and introduces a concrete, RL-compatible shaping that pushes models toward judicious tool use without hand-crafted rules or heavy SFT. The result is a simple, plug-and-play recipe that system builders can directly try in search and code agents.

**Strengths:**

The multiplicative tool-efficiency factor is easy to add to existing RL pipelines, and the GRPO variant provides a data-driven estimate of the “optimal” call budget per question. This is engineering-friendly and aligns with system constraints.

The paper shows large TC reductions and strong TP gains on both search and code tool settings, including OOD QA, while preserving accuracy on larger models. This is the right way to argue for deployment.

The case study and the “reasoning versus acting” analysis are insightful, demonstrating that discouraging gratuitous tools increases internal reasoning, which is exactly the intended effect.

**Weaknesses:**

EM and TP are informative, but many decisions hinge on end-to-end cost. Please add wall-clock latency, prompt token counts including tool observation length, and number of forward passes so that compute is matched across baselines. Some comparisons could still be explained by budget differences.

The reward uses alpha and a smooth constant c. Provide ablations showing stability, convergence speed, and EM-TP trade-offs as these vary, ideally on two datasets and two backbones. The text hints at PPO versus GRPO stability but does not quantify failure rates or variance.

The “minimal calls among correct trajectories” is a moving target. Analyze how often the proxy changes during training, whether it biases toward under-search, and how robust it is to spurious lucky trajectories. A small controlled synthetic task where the true optimum is known would help.

**Questions:**

(1) Under equalized prompt tokens, number of model calls, and observation lengths, how do OTC-PPO and OTC-GRPO compare to Search-R1 and ToRL on EM and TP, including wall-clock? A clean, matched table would increase confidence that gains are not budget artifacts.

(2) Show how often the estimated n changes per question during training, and the effect on stability. Include a diagnostic on under-search errors created by too-aggressive penalties.

(3) Can you demonstrate the same shaping works with larger tool menus or multi-tool plans, for example search, code, and calculator together, where the notion of “optimal calls” may be more complex. A small ablation would help.

**Details Of Ethics Concerns:**

The work uses public benchmarks, releases code snippets for reward design, and does not process sensitive personal data or human subjects.

---

> ### Author Response · Authors · 2025-11-24
>
> Thank you for the positive comments and acknowledgement of our originality and significance in RL pipeline of agents. Regarding of the comments and questions, we provide detailed response below.
>
> Weakness:
>
> > W1: end-to-end cost analysis
>
> -   We appreciate the reviewer’s focus on end-to-end cost. We would like to clarify a potential misunderstanding: **OTC-PO and all baselines does not set or limit any compute budget.** Therefore, there are no predefined constraints on inference steps, number of tool calls (despite that we have set the same maximal tool tools for all methods), forward passes, or reasoning/token length for all methods. In sepcific, our objective is to reduce unnecessary tool calls, thereby reducing real compute cost indirectly without imposing explicit budgets. Compared with the baseline, we mainly change the reward function and slightly change the prompt without any other changes.
>     Furthermore, we detailed track the number of tool calls, running times of different methods during training as shown in Figure 2. It is evident that our proposed method can reduce the number of tool calls and significantly reduce the running time, being more efficient during training. We also report the average tool call and tool productivity with EM during inference at main tables. Therefore, we believe our comparisons is fair enough to showcase the effectiveness and efficiency of our proposed method, and the performance gains are definitely **not budget artifacts.**
>
> > W2: ablation studies
>
> -   We provide detailed ablation study about the alpha and the constant big C at Appendix F and G. The smooth constant c is a small positive number and has very limited effects on the reward score. Therefore, our proposed method is not sensitive to this parameter. Regarding PPO vs. GRPO stability. We fully agree that this is an active research direction. Our goal is not to benchmark optimization algorithms, but to show that OTC-PO is compatible with both and remains effective under either training scheme. We additionally report variance across multiple runs to provide a quantitative view of stability, without drawing conclusions tied to specific optimization algorithms, as such analysis is beyond the scope of this work.
>
>     | Methods | NQ-EM(&uarr;) | NE-TC(&darr;) | NQ-TP(&uarr;) | HotpotQA-EM(&uarr;) | HotpotQA-TC(&darr;) | HotpotQA-TP(&uarr;) |
>     |------|------|------|------|------|------|------|
>     | OTC-PPO-3B    |   0.375&plusmn;0.028  |  0.995&plusmn;0.021    |   0.377&plusmn;0.037   |  0.268&plusmn;0.011    |  1.040&plusmn;0.020    |   0.258&plusmn;0.006   |
>     |  OTC-PPO-7B    |  0.451&plusmn;0.006    |  1.074&plusmn;0.048    |  0.420&plusmn;0.013    |  0.386&plusmn;0.004    | 1.410&plusmn;0.076     |   0.274&plusmn;0.017   |
>
> > W3: other analysis
>
> -   Yes, we track the changes of minimal tool calls  $n$ among correct trajectories as required. It is important to note before that the minimal tool call $n$ per question in our setting will naturally decrease or keep same during training since we update the global minimum using local minimum (as explained in line 213-215 and Figure 1) if the policy model find correct solutions with less tool calls. During training, we find $n$ in about 74% samples deccreases, and about $n$ in 26% samples keeps the same and n is 0 for more than 70% samples in these cases. The maximum drop is 3, and the average decrease is 1.8.

---

> ### Author Response · Authors · 2025-11-24
>
> Questions:
>
> 1. Please refer to weakness 1.
>
> 2. Please refer to weakness 3 for training analysis. During inference, we also investigate the under-search errors by providing the detailed analysis in Table 3. It is evident that our proposed method mostly delivers more efficient behaviors with the same answer with baseline (i.e., more than 80% ME in 7B model), and our proposed method delivers worse answer in less than 12\% cases (i.e., LA in table 3), mostly about 5%. Out of LA cases, 95\% is caused by under-search. In summary, **the under-search cases occupy less than 10\% in the 7B model, less than 5\% in the 3B model.**
>
> 3. Thank you for the suggestion, we extend our reward design to more challanging GAIA and HLE setting following the paper [1], we use the same wiki doc in Search-R1 for fair comparsion, and the results are shown as follows:
>
>     | Methods | GAIA-EM | GAIA-TC | GAIA-TP | HLE-EM | HLE-TC | HLE-TP |
>     |------|------|------|------|------|------|------|
>     |  Search-R1-PPO    |   7.1   |  6.2    |   1.1   |  3.3    |  3.2   |   1.0   |
>     |  OTC-PPO   | 10.7    | 0.9     |  11.9    |  4.5    |  0.4    | 11.3     |
>
>     It shows that our proposed method can naturally extend to these multi-tool involved situations and achieves even much better performance on tool call and tool productivity. We attribute this to more serious cognitive offloading issues in these complex settings.
>
>
> [1] Agentic Reinforced Policy Optimization

---

### Official Review · Reviewer_d63o · 2025-10-28

**Soundness:** 3
**Presentation:** 2
**Contribution:** 2
**Rating:** 4
**Confidence:** 4

**Summary:**

This paper proposes Optimized Tool Call-controlled Policy Optimization (OTC-PO), a RL framework that trains language models to use external tools more efficiently while maintaining accuracy. It introduces a new reward design and metric, tool productivity, which balances correctness with the number of tool calls. Experiments on multiple benchmarks with Qwen models show that OTC-PO reduces tool calls by up to 68% and improves tool productivity by over 200%, without significant loss in performance.

**Strengths:**

- OTC-PO can be plugged into existing RL pipelines with minimal modification. Its reward design is lightweight yet conceptually impactful.
- Evaluated on multiple tasks (search and code), datasets, and model scales (3B–7B), showing consistent improvements.

**Weaknesses:**

- The main novelty lies in the reward design, specifically the cosine-decay modulation to penalize excessive tool use. While effective, this idea feels heuristic and somewhat ad hoc. It would be stronger if compared with or generalized to other decay strategies (e.g., linear, exponential, adaptive).
- The paper lacks comparisons with existing length-control methods such as Length-Control Policy Optimization (LCPO) or budgeted reasoning RL approaches.
-

**Questions:**

- Could the authors justify why cosine decay is preferable to simpler alternatives (e.g., linear, exponential, or adaptive decay)? Including ablations with different decay types or visualizations of reward curves would help clarify robustness.
- In Section D, why do the authors revise the system prompt of Search-R1 by adding the instruction “You need to make every search call count and gain helpful results”? What motivates this modification, and does it influence the accuracy, training stability, or comparability of baselines?
- In Table 1, OTC-GPRO shows huge improvement over Search-R1-GRPO. Could the authors explain it?

---

> ### Author Response · Authors · 2025-11-24
>
> Thank you for your detailed review and we would like to make some important clarifications and address your concern throughly.
>
> Weakness:
>
> > W1: role of cosine-decay modulation in our reward design
>
> - We appreciate the reviewer’s question regarding the cosine-decay component. We would like to clarify that the cosine shape itself is not the core novelty. **It is only one implementation detail within a broader design framework** whose goal is two-fold:
>
>     - **Penalize unnecessary tool calls in a task– and model–adaptive way**. Unlike simple decay functions, this requires estimating the minimal correct tool call for each question and each model, which is nontrivial because the "optimal" calls is jointly determined by model ability and task difficulty. OTC-PO leverages this empirical optimal budget and then applies a decay function (cosine in our case) to rank correct trajectories. Other monotonic functions (e.g., linear or exponential decay) could also be used here, and we explicitly note this flexibility (line 204-206). **We chose cosine because it is smoother and easier to tune in practice, a property also discussed in recent reward-shaping work (e.g., [1]). The decay plots are provided in Figure 4 (Appx. D.1).**
>
>     - **Preserve accuracy while avoiding reward hacking**. Decay alone is not sufficient. The second part of our design, multiplying the efficiency reward with correctness, ensures that models only receive efficiency incentives after producing a correct answer. This correctness-gated structure prevents reward hacking and is formally analyzed in Lines 230–248 and we also provide theoretical justification at Appendix E.2.
>       **In summary, cosine is a tunable choice inside an adaptive, correctness-conditioned framework. The key contribution is not the decay curve itself, but learning the optimal tool calls from successful trajectories while preserving correctness.**
>
>     - We also try to implement the linear and exponential reward functions for OTC-PPO on QWen-3B. Specifically, after mapping m to [0, pi / 2), we then adopt f(x) = - (2/pi)*x + 1 (linear one); and f(x) = e^(-x) (exponential one), the results can be found below:
>
>         | Methods | NQ-EM(&uarr;) | NE-TC(&darr;) | NQ-TP(&uarr;) | HotpotQA-EM(&uarr;) | HotpotQA-TC(&darr;) | HotpotQA-TP(&uarr;) |
>         |------|------|------|------|------|------|------|
>         |  Search-R1-PPO    |   0.403   |  1.738    |   0.232   |  0.279    |  1.716    |   0.163   |
>         |  OTC-PPO-linear    | 0.348     |  1.055     |  0.330    |  0.252    |  1.109    | 0.227     |
>         |  OTC-PPO-exponential    |  0.362    |  1.140    |  0.317    |  0.256    | 1.205     |   0.212   |
>         |  OTC-PPO-cosine (original)    |  0.355    |  1.010    |  0.351    |  0.260    | 1.026     |   0.253   |
>
>
>         It confirms that our used cosine is slightly better than other choices, but more importantly, **all choices of different monotonic functions can reduce the tool call and increase the tool producitivity, since the behavior is majorly determined by our final reward design as explained above.**
>
> > W2: comparisons with existing length-control methods
>
> - We thank the reviewer for suggesting a comparison with LCPO and budgeted reasoning RL. However, **these approaches target a fundamentally different problem from OTC-PO, making a direct comparison misleading rather than informative.** The key distinctions are:
>
>     - **Different target behavior (acting vs. reasoning)**: Length-control and budgeted reasoning methods constrain the reasoning tokens. OTC-PO instead regulates external actions --tool calls -- which are discrete, high-cost operations. In essence, length methods promote "shorter reasoning", while OTC-PO learns "acting only when necessary". Our work specifically tackles cognitive offloading, the tendency of agentic LLMs to over-rely on tools rather than leveraging internal reasoning (e.g., generating better search queries or directly answer the question). Such overuse unnecessarily increases computation and infrastructure cost (see Fig. 3), which length-control methods do not address.
>
>     - **Different optimization objective (efficiency only after correctness)**: Most of length penalties discourage verbosity regardless of answer quality. OTC-PO minimizes tool usage only when the task is solved correctly (as advocated in our task definition). Inefficient but correct trajectories receive lower rewards than optimal correct ones, while all incorrect trajectories receive zero regardless of tool use. This correctness-conditioned optimization cannot be reproduced by generic length penalties, even dynamic ones.

---

> ### Author Response · Authors · 2025-11-24
>
> - **Different mechanism and implementations**: OTC-PO does not predefine a budget or cost for tool use. It learns empirical optimal tool usage by tracking the minimal correct calls per question and per model during training. The penalty then adapts as the model discovers better strategies. In contrast, LCPO and similar methods fix a pre-specified budget (e.g., using ngoldn_{\text{gold}}ngold in their paper), which cannot adapt to varying task difficulty or model capabilities.
>
> -  **Direct comparison**: To further address your concern, we run the latest reasoning length penalties algorithms [2] which just is accpted by NeurIPS 2025 on QWen-3B to provide direct comparison. We remove the tool results from the response and use the remaining content to calculate the reasoning length. The corresponding results are shown below.  The result aligns with our previous statements: 1) solely reasoning-conditioned algorithms have limited effects on the number of tool calls; 2) minimizing the number of reasoning tokens may decrease general performance in this setting.
>
>     | Methods | NQ-EM(&uarr;) | NE-TC(&darr;) | NQ-TP(&uarr;) | HotpotQA-EM(&uarr;) | HotpotQA-TC(&darr;) | HotpotQA-TP(&uarr;) |
>     |------|------|------|------|------|------|------|
>     | Reasoning-PPO    |   0.248   |  1.338    |   0.185   |  0.166    |  1.562    |   0.106   |
>     |  OTC-PPO   |  0.355    |  1.010    |  0.351 (+89.7%)   |  0.260    | 1.026    |   0.253 (+138.7%)   |
>
> In summary, these differences mean that length-penalty and budgeted reasoning methods do not apply to our setting. Instead, OTC-PO identifies new problem (a.k.a., cognitive offloading issues), and introduces both a new task definition and a correctness-conditioned, model-adaptive optimization framework to mitigate the problem, pointing to a new research direction beyond token-length control.
>
>   Questions
>
> - Please refer to weakness 1
>
> - We clarify that **we do not modify the original system prompts used in the baselines**, ensuring a fair and consistent comparison. We only replace one sentence "You can search as many times as you want." in the system prompt of original Search-R1 into "You need to make every search call count and gain helpful results" in our setting (as explained in line 304-305). We did this due to different targeted behavior and problems (as mentioned above), since we do not want the model to call tools as many times as it wants and we want it to call tools only when necessary.
>
> - Thank you for pointing this out. We believe it is another piece of evidence to show the great potential of our proposed method for the following reasons: 1) OTC-GRPO has a more accurate global approximation of the number of optimal tool calls required compared with OTC-PPO, since GRPO has a group of rollouts for each question (as explained in line 213-215). 2) The group design in GRPO provides clearer reward signals that better amplify the desired behavior, as shown in Figure 3.
>
>
> Thank you again for the valuable suggestions and comments, we will incoporate these analysis into our next version.
>
> [1] Demystifying long chain-of-thought reasoning in llms, ICML 2025
>
> [2] Training Language Models to Reason Efficiently, NeurIPS 2025

---

### Official Review · Reviewer_nebb · 2025-10-31

**Soundness:** 3
**Presentation:** 3
**Contribution:** 2
**Rating:** 4
**Confidence:** 4

**Summary:**

This paper introduces Optimized Tool Call-controlled Policy Optimization (OTC-PO), a simple yet effective RL-based framework that encourages models to produce accurate answers with less tool calls. Tool integrated reasoning is framed as jointly optimizing both correctness and efficiency by modulating the reward signal with a scaling coefficient that reflects tool efficiency. Two instantiations of the framework, OTC-PPO and OTC-GRPO, are extensively evaluated on two tools namely search and code to validate the effectiveness.

**Strengths:**

- The paper is generally clear and well structured. Figure 1 is a good illustration of the proposed method.
- The proposed method is simple yet effective and is compatible with various RL methods.
- Conducted experiments validate the effectiveness in reducing total tool calls with minimal to no loss of efficacy.

**Weaknesses:**

- The technical contribution of OTC-PO centers on scaling the correctness reward by a tool-efficiency coefficient that penalizes trajectories with a larger number of tool calls. While the prompt-adaptive estimation of minimal tool usage is a useful heuristic, conceptually it is an incremental extension of standard length penalties and reward shaping techniques for tool-efficiency.

- A follow-up to Search-R1 [1] demonstrates that incorporating a format reward can substantially improve performance, particularly when training from base LLMs. For example, using Qwen2.5-3B-Base, Search-R1 with a format reward reaches **0.428** (NQ) and **0.371** (HotpotQA) under PPO, and **0.429** (NQ) and **0.372** (HotpotQA) under GRPO. With Qwen2.5-7B-Base, the PPO variant attains **0.488** (NQ) and **0.436** (HotpotQA), while the GRPO variant achieves **0.458** (NQ) and **0.412** (HotpotQA); refer to Table 1 in [1]. These results are competitive with, and in some settings exceed, the performance of OTC-PO variants. A more extensive comparison against such methods would better contextualize the contribution.

- The paper reports single-run results and notes that hyper-parameters are reused from Search-R1. However, LLMs are known to exhibit high run-to-run variance, and sampling parameters can significantly influence outcomes. Without repeated trials and confidence intervals, it is difficult to assess the stability and practical significance of the reported improvements.

---

References

 [1] Jin et al., An Empirical Study on Reinforcement Learning for Reasoning-Search Interleaved LLM Agents

**Questions:**

1. Could the authors clarify the sampling parameters used during inference for all reported experiments? Such details are important for reproducibility and for understanding potential performance sensitivity to decoding settings.

2. Minor typos.  Line 106: “Only few of studies” should be revised to “Only a few studies.” Line 157: “generate outputs consists” should be revised to “the generated output, consisting.”

---

> ### Author Response · Authors · 2025-11-24
>
> Thank you for your detailed feedback on our paper and for raising valuable questions. Regarding all your concerns, we provide detailed response as follows:
>
>   Weakness:
>
> > W1:  incremental extension of standard length penalties
>
>   - We acknowledge that OTC-PO is related to reward shaping, and we cite length-penalty studies properly in our paper. However, **our contribution is not an incremental extension of length penalties. We clarify the distinction in three fundamental aspects:**
>
>     - **Different target behavior (acting vs. reasoning)**: Length-control and budgeted reasoning methods constrain the reasoning tokens. OTC-PO instead regulates external actions --tool calls -- which are discrete, high-cost operations. In essence, length methods promote "shorter reasoning", while OTC-PO learns "acting only when necessary". Our work specifically tackles cognitive offloading, the tendency of agentic LLMs to over-rely on tools rather than leveraging internal reasoning (e.g., generating better search queries or directly answer the question). Such overuse unnecessarily increases computation and infrastructure cost (see Fig. 3), which length-control methods do not address.
>     - **Different optimization objective (efficiency only after correctness)**: Most of length penalties discourage verbosity regardless of answer quality. OTC-PO minimizes tool usage only when the task is solved correctly (as advocated in our task definition). Inefficient but correct trajectories receive lower rewards than optimal correct ones, while all incorrect trajectories receive zero regardless of tool use. This correctness-conditioned optimization cannot be reproduced by generic length penalties, even dynamic ones.
>     - **Different mechanism and implementations**: OTC-PO does not predefine a budget or cost for tool use. It learns empirical optimal tool usage by tracking the minimal correct calls per question and per model during training. The penalty then adapts as the model discovers better strategies. In contrast, Length-Control Policy Optimization (LCPO) and similar length-based methods fix a pre-specified budget (e.g., using ngoldn_{\text{gold}}ngold in LCPO paper), which cannot adapt to varying task difficulty or model capabilities.
>      -   **Direct comparison**: To further address your concern, we run the latest reasoning length penalties algorithms [1] which is just accepted by NeurIPS 2025 on QWen-3B to provide direct comparison. We remove the tool results from the response and use the remaining content to calculate the reasoning length. The corresponding results are shown below.  The result aligns with our previous statements: 1) solely reasoning-conditioned algorithms have limited effects on the number of tool calls; 2) minimizing the number of reasoning tokens may decrease general performance in this setting.
>
>
>     | Methods | NQ-EM(&uarr;) | NE-TC(&darr;) | NQ-TP(&uarr;) | HotpotQA-EM(&uarr;) | HotpotQA-TC(&darr;) | HotpotQA-TP(&uarr;) |
>     |------|------|------|------|------|------|------|
>     | Reasoning-PPO    |   0.248   |  1.338    |   0.185   |  0.166    |  1.562    |   0.106   |
>     |  OTC-PPO   |  0.355    |  1.010    |  0.351 (+89.7%)   |  0.260    | 1.026    |   0.253 (+138.7%)   |
>
>     In summary, these differences mean that length-penalty and budgeted reasoning methods do not apply to our setting. Instead, **OTC-PO identifies new problem (a.k.a., cognitive offloading issues), and introduces both a new task definition and a correctness-conditioned, model-adaptive optimization framework to mitigate the problem, pointing to a new research direction beyond token-length control.**
>
> > W2: compasion with formard-incorporated reward design
>
>   - Thank you for the suggestion. First of all, as mentioned in our paper (lines 247-248), it is important to emphasize that our proposed OTC-PO reward is fully compatible with incorporating a format reward since this does not change the theoretical analysis or the desired behavior of our method, as detailed discussed in Appendix E.3 (line 782-791). Secondly, for completeness, we re-run the baseline and our proposed method on Qwen-3B with format constraints while keeping other settings same, **the results below confirm 1) simply format reward has limited effects on the number of tool calls; 2) our OTC-PO continues to reduce tool overuse under this extended reward.**
>
>     | Methods | NQ-EM(&uarr;) | NE-TC(&darr;) | NQ-TP(&uarr;) | HotpotQA-EM(&uarr;) | HotpotQA-TC(&darr;) | HotpotQA-TP(&uarr;) |
>     |------|------|------|------|------|------|------|
>     |  Search-R1-PPO    |   0.403   |  1.738    |   0.232   |  0.279    |  1.716    |   0.163   |
>     |  Search-R1-PPO-format    | 0.425     | 1.762     |  0.241    |  0.369    |  1.705    | 0.216     |
>     |  OTC-PPO-format    |  0.408    |  1.051    |  0.388 (+61.0%)    |  0.346    | 1.098     |   0.315 (+45.8%)  |

---

> ### Author Response · Authors · 2025-11-24
>
> > W3: significance of our proposed method
>
> -   We acknowledge the value of multiple runs and statistical validations. However, it is important to point out 1) **RL training is extremely computation-intensive, and similar to prior work, our initial submission reports single-run results under consistent conditions.** We  conduct additional runs for 3B and 7B models using our proposed method, and the results below confirm the robustness of our method. 2) **Our performance on tool call and tool productivity (TP) are significantly better than baselines, and the performance gap is large and consistent across datasets and model scales** (as evidenced in both search and code situations), indicating that our method’s efficiency benefits are robust, even under potential variability.
>
>     | Methods | NQ-EM(&uarr;) | NE-TC(&darr;) | NQ-TP(&uarr;) | HotpotQA-EM(&uarr;) | HotpotQA-TC(&darr;) | HotpotQA-TP(&uarr;) |
>     |------|------|------|------|------|------|------|
>     |  Search-R1-PPO-3B    |   0.403   |  1.738    |   0.232   |  0.279    |  1.716    |   0.163   |
>     | OTC-PPO-3B    |   0.375&plusmn;0.028  |  0.995&plusmn;0.021    |   0.377&plusmn;0.037   |  0.268&plusmn;0.011    |   1.040&plusmn;0.020    |   0.258&plusmn;0.006   |
>     |  Search-R1-PPO-7B    |   0.449   |  3.282    |   0.136   |  0.380    |  3.741    |   0.101   |
>     |  OTC-PPO-7B    |  0.451&plusmn;0.006    |  1.074&plusmn;0.048    |  0.420&plusmn;0.013    |  0.386&plusmn;0.004    | 1.410&plusmn;0.076     |   0.274&plusmn;0.017   |
>
>
> Questions:
>   1. We use the same training settings as ToRL and Search-R1, and apply greedy decoding during inference to avoid randomness.
>   2. We already fixed these minor typos in the revised version.
>
> [1] Training Language Models to Reason Efficiently, NeurIPS 2025
>
> Thank you again for detailed review and suggestions, we will incorporate these analysis in our next version.

---

### Author Response · Authors · 2025-11-28

Dear Area Chair and all Reviewers,

We sincerely appreciate the time and effort you have devoted to reviewing our paper. Following suggestions of all reviewers, we have added additional results/write-ups that further support our core claim and responded point by point to each of your concerns. **As the discussion period between reviewers and authors is drawing to a close, we would be very grateful to know whether our responses have adequately addressed your concerns; we are, of course, happy to provide any further clarification.**

Thank you again for your time and efforts. Looking forward to your feedback.

Best Wishes

All authors

---

### Author Response · Authors · 2025-12-01
**Rebuttal Summary**

Dear Reviewers, AC, SAC and PC,

We sincerely thank you for your time, efforts, and careful evaluation of our manuscript. **Unfortunately, the reviewers do not actively partipate in the discussion after we provided our detailed response, and later due to an unexpected and well-known incident, the reviewers were unable to engage in further discussion.** We regret losing the opportunity to continue the dialogue and clarify our work further.  Nevertheless, we would like to emphasize that our response includes thorough explanations and comprehensive evidence, and we firmly believe that it adequately addresses all of the reviewers' concerns.

To facilitate your quick understanding of the paper and our rebuttal, we provide a concise summary below. Briefly, in this paper, we propose Optimized Tool Call-controlled Policy Optimization (OTC-PO), a RL framework that trains language models to optimize both answer correctness and efficiency of tool use. It introduces a new reward design and metric, tool productivity, which balances correctness with the number of tool calls.

**Key strengths highlighted by reviewers**
  - **The paper is clear, well motivated and structured**. Reviewers agreed that our paper is well written and easy to understand, the figures are good illustrations and analysis is insightful. (`Reviewer nebb`, `Reviewer y9uq`, `Reviewer oxjH`)
  - **Our method is simple, effective, and easily pluggable with existing RL methods.** Reviewer considered our proposed method is compatible with various RL methods with minimal modification. The reward design to minimize the number of tool calls is lightweight yet conceptually impactful, and it is engineering-friendly and aligns with system constraints. (`Reviewer nebb`, `Reviewer d63o`, `Reviewer y9uq`).
  - **Extensive experiments validate the effectiveness of reducing total tool calls with minimal to no loss of efficacy.** All reviewers agreed that our experiments are extensive in terms of tasks (search and code), datasets, and model sizes. Our results showcase consistent improvements and are exactly the intended effect. (`Reviewer nebb`, `Reviewer d63o`, `Reviewer y9uq`, `Reviewer oxjH`)

**Key revisions in response to the comments**

  - [`Reviewer nebb` and `d63o`] **Difference and comparison with standard length penalties algorithms.** We clarified that our method is significantly different from existing reasoning length-based RL methods in terms of targeted problem / behavior, optimization objective and implementations. Further, we also add direct comparison with one of most cited length algorithms, and the results reconfirm the superiority of our proposed method.
  - [`Reviewer nebb`, `d63o`, `y9uq`] **Ablation Study.** The reviewers requested additional ablation studies, including analyses of the format reward, comparisons with alternative reward functions, and evaluations using larger model sizes (≥14B). We provide all of them in response, and the results align with our main observation and statements.
  - **Other minor clarifications and questions.** We provide detailed analysis and explanation as requested by each reviewers.

We sincerely appreciate the constructive feedback and would like to emphasize that we have addressed all raised concerns in the responses as below (feel free to check details in our responses). We respectfully ask the ACs, SACs, and PCs to consider our detailed responses and **estimate how the reviewer's impressions would have changed had the discussion period not been cut short.** Once again, we’re disappointed to lose the opportunity and it is not fair to papers of our case, but we fully respect the code of conduct and the policies.

Thank you again for your time, thoughtful evaluation. We truly appreciate your efforts and hope this summary is helpful.

Respectfully and sincerely,

All authors

---

### Meta-Review · Area_Chair_PNry · 2025-12-25

**Summary:**

This submission studies efficiency in tool-integrated reasoning (TIR): rather than optimizing only final correctness, the authors aim to reduce unnecessary tool calls (and thus latency/cost) while maintaining accuracy. They propose Optimized Tool Call-controlled Policy Optimization (OTC-PO), a reinforcement-learning framework that adds a correctness-gated tool-efficiency factor to the reward, and introduce tool productivity (correct answers per total tool calls) as an evaluation metric.

The framework is instantiated as OTC-PPO and OTC-GRPO and evaluated across search- and code-based QA benchmarks with Qwen-family models. Reviewers agree the paper is generally clear and the empirical results demonstrate substantial reductions in tool calls and strong gains in tool productivity, with limited degradation in accuracy at larger scales. Overall the paper is based on some empirical obsersations that the number of tool calls should be small thus design an auxuiliary reward term to encourage efficient tool call usages. However the motiviation is often not correct in the realistic scenarios such as deep research or SWE tasks. I think the design is narrow and thus learn to reject the paper.

**Reviewer Concerns:**

Across the four reviews, the main points of discussion were:

- Novelty / relation to prior reward shaping and length-control work. Two reviewers viewed the core idea as close to standard reward shaping / “length penalties,” and requested clearer positioning and/or direct comparisons; relatedly, one reviewer asked for comparisons to length-control/budgeted reasoning approaches and alternative decay strategies.

- Compute / cost accounting and robustness. One reviewer requested more end-to-end cost evidence (e.g., wall-clock latency, prompt/observation length, forward passes) and cautioned that some gains might be artifacts of mismatched budgets; another asked for multiple runs / variance given RL instability.

- Stability of the “minimal tool calls among correct trajectories” proxy and risk of under-search. A reviewer asked how often this proxy changes and whether it biases toward under-search or exploits “lucky” trajectories; they also suggested controlled diagnostics.

- Protocol details / fairness / missing related work. Questions included inference decoding settings, prompt changes relative to baselines, and a missing related work citation.

The rebuttal is substantive and addresses many of the above:

- The authors clarify that their target is external actions (tool calls) rather than reasoning length, and emphasize a correctness-gated efficiency incentive; they also provide ablations over alternative decay choices (linear/exponential vs cosine) and argue cosine is not the key novelty.

- They add robustness evidence via additional runs (reporting mean±std for 3B/7B) and specify greedy decoding at inference to reduce randomness.

- They provide diagnostics related to under-search, reporting that under-search cases are a minority under their analysis, and extend experiments to more complex settings (GAIA/HLE), suggesting the reward generalizes beyond the simplest tool setups.

- They respond to practical-cost concerns with time analysis (reported training-time reductions and some inference-time reduction) and add a 14B result to support scalability.

Remaining limitations are mostly about standardization/normalization of efficiency evidence: while the rebuttal includes timing and tool-call statistics, the paper still does not fully provide compute-matched, end-to-end cost tables (tokens incl. tool observations, model forward passes, latency under a fixed serving setup) that would make “cost” claims maximally airtight.

**Reviewer Scores:**

I cannot reliably answer this counterfactual question without putting words in reviewers’ mouths. I will not impute score changes beyond what reviewers explicitly stated in the discussion. I instead provide a faithful synthesis of the discussion outcomes and remaining points of disagreement.

---

### Decision · Program_Chairs · 2026-01-26

Reject